# Cross-Tactile Sensor Representation Learning

**Yan Zhang**[1]   **Zheng Wang**[1]   **Pengpeng Zeng**[1]   **Xing Xu**[1]   **Jingkuan Song**[1]   **Heng Tao Shen**[1]

## Abstract

Visuo-tactile sensors have been widely adopted in robotic manipulation. However, inherent heterogeneity in sensor designs hinders the learning of unified tactile representations in cross-sensor scenarios. Existing methods that focus on reconstruction or task-specific supervision often fail to capture the common information between different tactile sensors, particularly in the presence of substantial sensor variations, resulting in limited generalization to unseen sensors. To address this, we propose Cross-Tactile Sensor Representation Learning (CTSRL), a unified framework for sensor-agnostic tactile representation learning. CTSRL introduces a Cross-Sensor Modulator (CSM) to eliminate sensor-specific biases and adopts a two-stage learning paradigm: (1) leveraging aligned synthetic data for cross-sensor self-supervised learning to extract shared latent representations across sensor domains; and (2) integrating real-world multimodal tactile data to bridge the sim-to-real semantic gap through cross-modal alignment, thereby enriching representations with fine-grained semantic attributes. Experimental results show that our method demonstrates strong multi-sensor generalization, significantly improving sensor-agnostic representation learning.

## 1. Introduction

In the robotics community, visuo-tactile sensors have garnered significant attention due to their ability to provide rich and detailed information regarding contact surfaces. Numerous studies have leveraged these sensors to achieve fine-grained operations, such as robotic grasping (Calandra et al., 2018) and fabric manipulation (Sunil et al., 2023). However, visuo-tactile sensors exhibit extreme heterogeneity due to the absence of unified design and sensing stan-

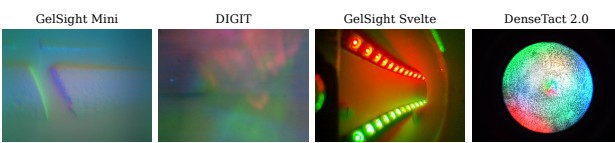

GelSight Mini    DIGIT    GelSight Svelte    DenseTact 2.0

*Figure 1.* Illustration of tactile sensor heterogeneity across different hardware configurations. Tactile images captured from different visuo-tactile sensors exhibit significant variations in geometry, illumination, and color distribution.

dards. As shown in Figure 1, sensors such as GelSight Mini (GelSight Inc., 2026), DIGIT (Lambeta et al., 2020), and DenseTact2.0 (Do et al., 2022) vary in their geometry, camera specifications, and illumination configurations. Consequently, the tactile images they produce can differ drastically even under identical contact conditions. This heterogeneity makes models trained on specific sensors difficult to apply directly to others. When encountering a new sensor, researchers often need to re-collect large amounts of data for retraining. Therefore, learning sensor-agnostic universal representations is of paramount importance in practice.

Various methods have been developed to enhance generalization to unseen sensor domains, either at the data or model level. Data-level approaches (Yuan et al., 2017; Calandra et al., 2017) attempted to improve generalization by using multiple GelSight sensors to gather diverse tactile datasets, but this approach provides only minor performance gains. Model-level methods such as T3 (Zhao et al., 2024) enhance transferability by pretraining a shared Transformer chunk with sensor-specific encoders and task-specific decoders. However, none of them explicitly disentangles sensor-invariant information from different tactile sensors. To this end, CTTP (Rodriguez et al., 2025) and SITR (Gupta et al., 2025) begin to combine contrastive learning (Chen et al., 2020) with aligned multi-sensor tactile data to learn transferable tactile representations for zero-shot object classification and pose estimation. Nevertheless, their downstream evaluations either omit transfer to unseen tactile sensors or strongly depend a on reconstruction decoder, degrading their claimed transferability. Furthermore, they do not further explore the fine-grained semantic capabilities of the learned representations, such as material identification, which are crucial for robotic manipulation like grasping (Jiang & Luo, 2022; Gu et al., 2025).

To address these challenges, we propose **C**ross-**T**actile

---

[1]School of Computer Science and Technology, Tongji University, China. Correspondence to: Zheng Wang <zh_wang@hotmail.com>.

*Proceedings of the 43^{rd} International Conference on Machine Learning*, Seoul, South Korea. PMLR 306, 2026. Copyright 2026 by the author(s).

Sensor Representation Learning (CTSRL), a novel framework designed to extract sensor-invariant representations from different tactile sensors. The architecture of CTSRL is centered around a learnable Cross-Sensor Modulator. Built upon this modulator, our approach employs a two-stage training paradigm: it first achieves structural alignment through cross-sensor representation learning, and subsequently performs semantic enrichment via multi-modal alignment.

Unlike traditional methods that enforce rigid feature constraints, CSM adaptively modulates tactile representations using sensor-aware embeddings, mapping features from different sensors into a shared latent space. We also incorporate a strategy that randomly replaces sensor-specific embeddings with a universal one during training (Feng et al., 2025), encouraging the model to leverage shared cross-sensor knowledge when encountering unseen sensors.

By leveraging CSM-modulated tactile features, we conduct Cross-sensor Representation Learning on aligned simulated data (Gupta et al., 2025). The primary objective is to guide the cross-sensor correlation matrix toward the target diagonal matrix, prompting the model to filter out hardware-specific artifacts and focus on shared physical invariants (Wang et al., 2026a). Additionally, we incorporate intra-sensor learning to prevent representation collapse, which strengthens cross-sensor alignment through enhanced intra-sensor knowledge. By combining cross-sensor and intra-sensor learning to align geometric features while using the CSM to remove sensor-specific variations, we ensure the development of robust, sensor-agnostic representations.

Furthermore, to bridge the sim-to-real domain gap, CTSRL employs real-world tactile-vision-language (T-V-L) data for Cross-modal Semantic Alignment (Wang et al., 2024b; Xu et al., 2026). While simulation provides structural consistency, it often lacks the intricate textures required for high-level tasks (Jianu et al., 2022). This multi-modal alignment enables the model to capture fine-grained physical properties—such as hardness and roughness—while further enhancing the generalization of tactile feature learning.

Our main contributions are as follows:

- We design a Cross-Sensor Modulator to adaptively mitigate inter-sensor biases, thereby capturing the shared latent representations between different sensors.

- We propose an effective cross-sensor representation learning method that extracts sensor-agnostic representations through synergistic cross-sensor and intra-sensor learning strategies.

- We introduce a cross-modal alignment phase to empower the simulation-pre-trained model in capturing nuanced semantic and physical attributes.

## 2. Related Works

**Visuo-tactile Representation Learning.** Tactile information from visuo-tactile sensors can be represented as images, making the application of computer vision models and algorithms a common practice in the tactile domain (Zandonati et al., 2023). Driven by the collection of large-scale tactile datasets and advancements in sensor hardware (Yang et al., 2022; Wan et al., 2024), research in tactile representation learning has made significant progress (Higuera et al., 2025). Previous studies have explored various methodologies to learn these representations (Wang et al., 2024a); a currently prevailing approach involves aligning or fusing tactile data with other modalities, particularly vision (Dave et al., 2024; Wu et al., 2025). Other techniques leverage task-specific supervised learning and masked autoencoders for training (Cao et al., 2023; Zhao et al., 2024).

However, these efforts often overlook the domain gaps between heterogeneous sensors and fail to explore universal cross-sensor tactile representations applicable to diverse tasks. T3 (Zhao et al., 2024) trained separate encoder-decoder pairs for distinct sensor-task combinations and utilized a shared transformer backbone to learn common tactile features; nevertheless, this approach depends on large datasets and is constrained by specific encoder architectures. Other methods apply contrastive learning within sensor modalities to facilitate transfer learning. For instance, CTTP (Rodriguez et al., 2025) proposed Contrastive Tactile-to-Tactile Pre-training to learn shared latent representations across sensors. However, its pre-training and downstream evaluations rely on data from the same sensor domain, raising doubts about its potential for cross-sensor transfer. Similarly, SITR (Gupta et al., 2025) enables zero-shot transfer across tactile sensors through a pre-training approach that combines supervised contrastive learning with reconstruction. However, its downstream performance is highly contingent on the reconstruction decoder rather than the extracted sensor-invariant features (Wang et al., 2026b). Moreover, the absence of evaluations on fine-grained semantic tasks in these methods limits the discriminative capabilities required for real-world tactile applications (Yu et al., 2025).

To address this, we propose a Cross-Sensor Modulator with a two-stage framework, where transferable representations are learned via cross-sensor self-supervision and further refined through multi-modal semantic alignment.

**Multimodal Alignment.** This paradigm serves as a cornerstone of cross-modal intelligence (Baltrušaitis et al., 2018; Gong et al., 2025), aiming to establish robust semantic correlations between diverse data modalities (Lou et al., 2025), including vision, language, and touch. By fusing complementary information across modalities, it not only enhances the model's comprehension and discriminative power in complex scenarios (Xie et al., 2025; Chen et al., 2025) but

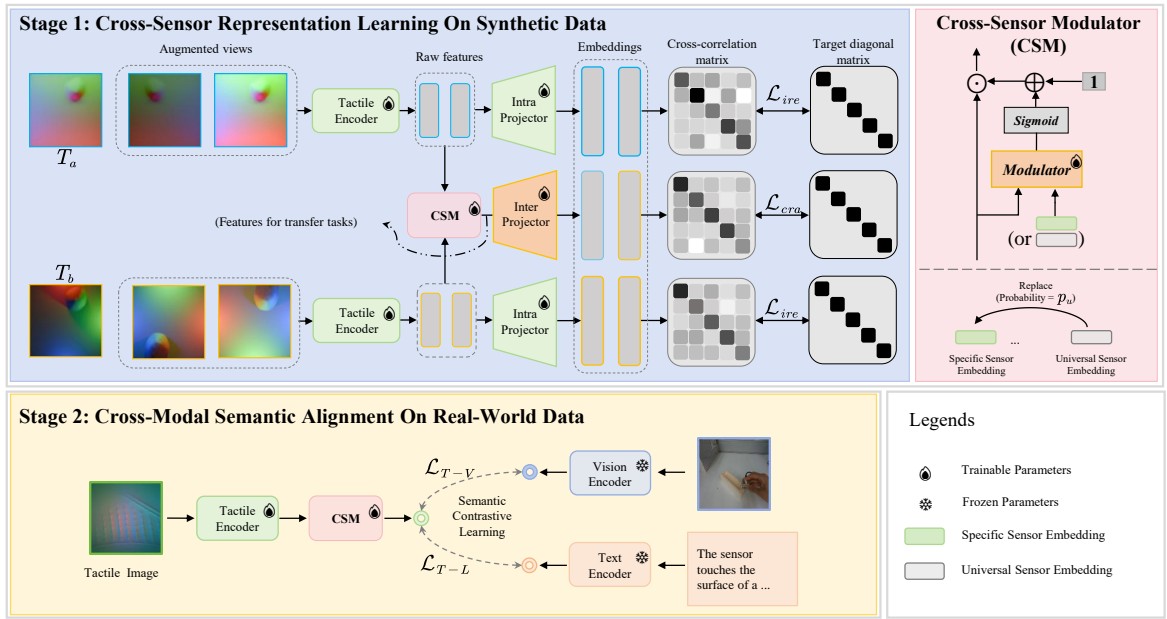

*Figure 2.* Overview of CTSRL, which consists of a Cross-Sensor Modulator (CSM) to remove sensor-specific biases and a two-stage framework. We first perform cross-sensor self-supervised learning on aligned synthetic datasets. The framework then leverages real-world multi-modal data to facilitate semantic alignment, bridging the gap between tactile perception and other modalities.

also indirectly bolsters representation learning within individual modalities (Ma et al., 2024). The advent of CLIP (Radford et al., 2021) marked a paradigm shift in the success of vision-language pre-training, subsequently leading to the integration of additional modalities including audio (Elizalde et al., 2023), video (Xu et al., 2021), and 3D point clouds (Zhang et al., 2022). Building on this trend, Image-Bind (Girdhar et al., 2023) constructed a unified embedding space across six modalities through image-pair learning, achieving remarkable performance in both visual and non-visual tasks. On this basis, UniTouch (Yang et al., 2024) inherits the pre-aligned visual-multimodal space from Image-Bind and extends it to the tactile domain. By unifying tactile signals with these pre-aligned modalities into a shared space, it enables robust performance across various zero-shot tactile applications. AnyTouch (Feng et al., 2025) employed a hierarchical architecture to fuse static and dynamic tactile data, leveraging a combination of masked modeling, multimodal alignment, and cross-sensor matching to learn unified representations.

Following this line of research, we harness the robust vision-language priors of CLIP to align with our pre-trained tactile encoder, aiming to strengthen the model's semantic comprehension of tactile information.

## 3. Method

In this section, we introduce our framework for CTSRL, as shown in Figure 2. The framework comprises a Cross-Sensor Modulator (CSM) and a two-stage training pipeline,

including cross-sensor self-supervised learning on simulated tactile data and cross-modal semantic alignment with real-world multimodal tactile data.

### 3.1. Cross-Sensor Modulator

We employ a ResNet-50 (He et al., 2016) as our tactile encoder $f_T$ (as shown in Table 1), mapping raw tactile inputs $T_s$ from sensor $s$ to a $d$-dimensional feature space. To facilitate training across multi-sensor tactile data, we introduce a CSM as a learnable calibrator during the feature extraction stage. This module assigns a corresponding $d_e$-dimensional embedding vector $e_k$ to each specific sensor (indexed by $k$), explicitly learning sensor-specific "style parameters" to dynamically modulate the shared feature space. Given a tactile feature $z = f_T(T_s) \in \mathbb{R}^d$, the CSM first compresses $z$ into a latent vector $z_r$, via a linear projection followed by a ReLU activation: $z_r = \text{ReLU}(W_r z + b_r) \in \mathbb{R}^{d_e}$, where $W_r \in \mathbb{R}^{d_e \times d}$. Subsequently, the CSM fuses $z_r$ with the sensor embedding $e_k$ through element-wise addition. The modulation coefficient $\alpha \in \mathbb{R}$ is then computed using a linear mapping and a sigmoid function: $\alpha = \sigma(W_e(z_r + e_k) + b_e) \in \mathbb{R}^d$, where $W_e \in \mathbb{R}^{d \times d_e}$. The final modulated feature $\tilde{z}$ is obtained as:

$$\tilde{z} = z \odot (1 + \alpha), \tag{1}$$

where $\odot$ denotes element-wise multiplication. Intuitively, CSM performs feature-wise modulation conditioned on both the input tactile response and the sensor identity. The intermediate feature $z_r$ captures sample-dependent sensitivity to sensor-induced variations, while the sensor embedding

*Table 1.* Statistical comparison of architectural configurations, parameter counts, and training scales against baseline models.

| Method | BaseModel | Param. | Tactile Training Data | Data Size |
|---|---|---|---|---|
| T3 (Zhao et al., 2024) | Custom-Based | 173.0M | FoTa Dataset (Zhao et al., 2024) | 3080k |
| CTTP (Rodriguez et al., 2025) | ResNet-50 | 25.6M | Paired Tactile Data | 62k |
| SITR (Gupta et al., 2025) | ViT-B | 96.0M | Synthetic Tactile Dataset (Gupta et al., 2025) | 1000k |
| UniTouch (Yang et al., 2024) | ViT-L | 304.9M | TAG, Feel, YCB-Slide, OF 2.0 | 490k |
| AnyTouch (Feng et al., 2025) | ViT-L | 304.0M | TAG, TVL, SSVTP, Octopi, TacQuad, VisGel, Cloth, YCB-Slide, OF Real | 2480k |
| AnyTouch$^+$ (Feng et al., 2025) | ViT-L | 304.0M | TAG, TVL, SSVTP, Octopi, TacQuad | 390k |
| **CTSRL (Stage 1)** | ResNet-50 | 26.1M | Synthetic Tactile Dataset (Gupta et al., 2025) | 50k |
| **CTSRL (Stage 2)** | ResNet-50 | 26.1M | TAG, TVL, SSVTP, Octopi, TacQuad | 390k |

$e_k$ encodes global, hardware-specific biases. Their combination predicts a channel-wise modulation coefficient that adaptively rescales $z$, enabling selective compensation of sensor biases while preserving semantic structure.

Motivated by Feng et al. (2025), we also incorporate a universal sensor embedding $e_u \in \mathbb{R}^{d_e}$. To enhance cross-sensor generalization, we employ a random substitution strategy during training to blend sensor-private information. This paradigm forces $e_u$ to learn a universal feature mapping logic that transcends individual sensor characteristics. During the training phase, we randomly replace the sensor-specific embedding $e_k$ with the universal one $e_u$ with probability $p_u$. The final embedding $e$ is subsequently applied to the calculation of the modulation coefficient $\alpha$. Settings of $p_u$ for different stages are provided in Section E.

### 3.2. Cross-Sensor Representation Learning

For each tactile sample, we randomly pair tactile images from two heterogeneous sensors, denoted as $a$ and $b$, to form a positive pair $(T_a, T_b)$. Each pair consists of tactile data captured from different simulated sensor configurations, but under the same or similar physical conditions, ensuring valid cross-sensor representation learning. Each tactile image in the pair is processed into two augmented views and fed into a shared tactile encoder, yielding latent features $z_a^{(v)}, z_b^{(v)} \in \mathbb{R}^d$, where $v \in \{1, 2\}$ denotes the augmentation index.

These latent features are then bifurcated into two parallel processing streams. In the intra-sensor stream, features are passed through intra-sensor projectors to preserve sensor-specific details, projecting into a **raw space** $h \in \mathbb{R}^D$. Concurrently, in the inter-sensor stream, the Cross-Sensor Modulator (CSM) first modulates the latent features to align cross-sensor data, which are then mapped by inter-sensor projectors into a **modulated space** $\tilde{h} \in \mathbb{R}^D$. This dual-projection mechanism effectively captures shared, sensor-invariant semantics in $\tilde{h}$ while maintaining the semantic integrity of individual sensor characteristics in $h$, thereby reducing hardware-specific biases. These embeddings undergo batch normalization to be mean-centered along the

batch dimension. Subsequently, the normalized embeddings are utilized to compute both intra-sensor and cross-sensor cross-correlation matrices for optimization.

**Normalized Barlow Twins Objective.** We adopt a normalized version of the Barlow Twins loss (Zbontar et al., 2021) as our core supervision. For two batches of embeddings $X, Y \in \mathbb{R}^{B \times D}$, the normalized objective is defined as:

$$\mathcal{L}_{\mathrm{nBT}}(X, Y) \triangleq \frac{1}{D} \left( \sum_i (1 - \mathcal{C}_{ii})^2 + \lambda \sum_i \sum_{j \neq i} \mathcal{C}_{ij}^2 \right), \tag{2}$$

where $\mathcal{C} \in \mathbb{R}^{D \times D}$ is the cross-correlation matrix computed along the batch dimension. The $1/D$ factor serves as a **normalization term**, ensuring numerical stability and invariance to the latent dimensionality $D$.

**Cross-sensor Representation Alignment.** Utilizing the features already modulated by the CSM, we apply the cross-sensor alignment loss to the resulting embeddings $\tilde{h}_a$ and $\tilde{h}_b$ from disparate sensors. By enforcing cross-correlation constraints on these modulated representations, the model further suppresses persistent hardware-specific biases that may remain after initial calibration. This alignment process compels the encoder to look past the unique "style" of individual sensors and concentrate on shared physical semantics, such as contact geometry and surface deformation. Consequently, this refinement ensures the extraction of sensor-invariant information, providing a robust foundation for tactile understanding across heterogeneous simulated configurations. The cross-sensor representation alignment loss is thus defined as:

$$\mathcal{L}_{\mathrm{cra}} = \mathcal{L}_{\mathrm{nBT}}(\tilde{h}_a, \tilde{h}_b), \tag{3}$$

where $\tilde{h}_a$ and $\tilde{h}_b$ are randomly sampled from their respective augmented views $\{\tilde{h}^{(1)}, \tilde{h}^{(2)}\}$.

However, directly enforcing cross-sensor alignment may lead to dimensional collapse, where the latent space fails to capture informative or discriminative features. To tackle this issue, we further introduce an intra-sensor representation enhancement within each sensor.

**Intra-sensor Representation Enhancing.** To prevent representation collapse and enhance the inherent discriminative power of individual sensors, we introduce an intra-sensor objective. This loss operates on the raw projected views $h^{(1)}$ and $h^{(2)}$ within the same sensor:

$$\mathcal{L}_{\text{ire}} = \mathcal{L}_{\text{nBT}}(h_a^{(1)}, h_a^{(2)}) + \mathcal{L}_{\text{nBT}}(h_b^{(1)}, h_b^{(2)}). \quad (4)$$

The overall training objective for the Cross-sensor Representation Learning (CRL) stage is a weighted sum of the two components:

$$\mathcal{L}_{\text{CRL}} = \lambda_{\text{cra}}\mathcal{L}_{\text{cra}} + \lambda_{\text{ire}}\mathcal{L}_{\text{ire}}, \quad (5)$$

where $\lambda_{\text{cra}}$ and $\lambda_{\text{ire}}$ are trade-off coefficients balancing cross-sensor alignment and intra-sensor feature preservation.

### 3.3. Cross-modal Semantic Alignment

While CRL stage establishes a sensor-agnostic foundational representation on simulated data, this approach possesses intrinsic limitations for real-world tasks. Specifically, the simulation-based methodology captures only macro-geometric deformations, neglecting critical micro-texture and friction characteristics, while also failing to provide high-level semantic grounding for abstract material properties (Jianu et al., 2022). To overcome these constraints, we introduce Cross-modal Semantic Alignment (CSA), which leverages real-world tactile-vision-language triplets for semantic contrastive learning.

Previous methods, such as AnyTouch (Feng et al., 2025), typically perform LoRA (Hu et al., 2022) fine-tuning on the CLIP vision encoder during tripartite alignment. While such fine-tuning enhances task-specific adaptation, it introduces a risk: disrupting the powerful prior knowledge of CLIP, which is crucial for multimodal alignment semantics. In light of this, we choose to freeze both the pre-trained CLIP vision and language encoders. We treat the resulting pre-aligned joint representation space as our anchors. This approach ensures that tactile features are calibrated under stable dual-modality constraints, facilitating the efficient extraction of fine-grained semantic information, including material properties and textures, from real-world inputs. Furthermore, by restricting the optimization solely to the tactile encoder and the CSM, this design promotes the learning of more discriminative and semantically meaningful tactile representations.

Given a batch of $B$ aligned vision-language-tactile triplets $\{(z_i^V, z_i^L, z_i^T)\}_{i=1}^B$, we align the tactile features with the pre-trained embeddings by maximizing the cosine similarity of corresponding pairs. We optimize this objective using the InfoNCE loss (Oord et al., 2018). Specifically, the tactile-language alignment loss $\mathcal{L}_{T-L}$ is formulated as:

$$\mathcal{L}_{T-L} = -\frac{1}{2B}\sum_{i=1}^{B}\left[\log\frac{\exp(\langle z_i^T, z_i^L\rangle/\tau)}{\sum_{j=1}^{B}\exp(\langle z_i^T, z_j^L\rangle/\tau)} \right.$$
$$\left. + \log\frac{\exp(\langle z_i^L, z_i^T\rangle/\tau)}{\sum_{j=1}^{B}\exp(\langle z_i^L, z_j^T\rangle/\tau)}\right], \quad (6)$$

where $\tau$ is a temperature hyperparameter. The first term aligns each tactile embedding $z_i^T$ with its paired linguistic embedding $z_i^L$, while the second term performs the symmetric alignment. The loss for vision-tactile pairs, $\mathcal{L}_{T-V}$, is defined analogously. By integrating these components, the multimodal alignment loss is given by:

$$\mathcal{L}_{\text{CSA}} = \mathcal{L}_{T-L} + \mathcal{L}_{T-V}. \quad (7)$$

By forcing the tactile encoder toward these frozen anchors, our approach not only enriches the representations with unified multimodal semantic attributes but also compensates for the lack of fine-grained semantic information in the simulated data. This process ultimately yields tactile representations with high semantic discriminative power and robust zero-shot generalization capabilities.

## 4. Experiments

In this section, we evaluate our framework in two stages: (1) Zero-shot cross-sensor transferability on geometric tasks (object classification and pose estimation). (2) Fine-grained semantic extraction capabilities (material classification) through multimodal alignment.

### 4.1. Zero-shot Cross-sensor Transfer

**Training Datasets.** For **Cross-sensor Representation Learning**, we utilize a subset of the synthetic tactile dataset from SITR (Gupta et al., 2025). This structured dataset provides the necessary geometric consistency to train our model, enabling it to effectively isolate shared physical invariants from hardware-specific biases. More details can be found in Section B.

**Benchmarks and Baselines.** We utilize real-world datasets from SITR (Gupta et al., 2025) and T3 (Zhao et al., 2024). The SITR benchmark comprises seven heterogeneous sensors—four GelSight Mini variants, Hex (Yuan et al., 2017), Wedge, and DIGIT—covering 16-class classification and pose estimation. Similarly, the T3 suite encompasses GelSight Mini, Wedge, Svelte (Zhao & Adelson, 2023), and DenseTact2.0 for 6-class object classification and pose estimation. We evaluate our method against several baselines, including standard ResNet-50 variants (with random initialization, ResNet RI, and ImageNet pre-training, ResNet PT) and specialized tactile transfer models such as T3, CTTP (Rodriguez et al., 2025), and SITR. Since T3

*Table 2.* Evaluation results for model transfer and no-transfer performance on the **SITR Bench**. **Intra-sensor** denotes transfer across GelSight Mini 1 to 4. **Inter-sensor** denotes transfer among GelSight Mini 1, Wedge, Hex, and DIGIT. **DIGIT-Mini**: Bidirectional transfer between DIGIT and GelSight Minis. We report the mean and standard deviation of transfer results among the sensor sets specified.

| Method | Object Classification (↑) | | | | Pose Estimation (↓) | | |
|---|---|---|---|---|---|---|---|
| | Intra-sensor | Inter-sensor | DIGIT-Mini | No transfer | Inter-sensor | DIGIT-Mini | No transfer |
| ResNet RI | 11.95 ±1.86 | 13.99 ±2.43 | 10.67 ±2.07 | 23.96 ±7.91 | 0.92 ±0.20 | 1.18 ±0.25 | 0.50 ±0.03 |
| ResNet PT | 75.44 ±17.55 | 73.03 ±14.58 | 50.31 ±11.05 | 96.48 ±3.94 | 0.81 ±0.19 | 0.92 ±0.10 | 0.48 ±0.02 |
| T3 | 38.66 ±20.63 | 17.02 ±8.55 | – | 93.77 ±2.87 | 1.70 ±0.07 | – | 0.51 ±0.02 |
| CTTP | 11.25 ±1.85 | 12.24 ±3.13 | 9.03 ±1.67 | 22.18 ±8.55 | 1.00 ±0.23 | 1.21 ±0.33 | 0.48 ±0.03 |
| SITR | 90.23 ±8.16 | 81.94 ±12.92 | 67.78 ±16.91 | 99.72 ±0.22 | 0.80 ±0.21 | 1.12 ±0.15 | 0.51 ±0.01 |
| **Ours (Stage 1)** | **95.00** ±4.62 | **88.52** ±8.01 | **80.62** ±7.45 | 99.58 ±0.45 | **0.75** ±0.16 | **0.79** ±0.04 | **0.45** ±0.06 |

*Table 3.* Evaluation of transfer results on the T3 Bench. W, S, M, and D denote Wedge, Svelte, Mini, and DenseTact2.0, respectively.

| Method | Object Classification (↑) | | Pose Estimation (↓) | |
|---|---|---|---|---|
| | W → S | M → D | W → S | M → D |
| ResNet RI | 15.26 | 17.66 | 9.14 | 3.89 |
| ResNet PT | 17.04 | 17.80 | 8.57 | 3.35 |
| T3 | 22.30 | 22.50 | 8.94 | 2.85 |
| CTTP | 17.04 | 23.02 | 8.77 | 2.89 |
| **Ours (Stage 1)** | **22.85** | **23.59** | **7.89** | **2.73** |

does not provide encoder weights for the GelSight Hex or DIGIT, we report its inter-sensor results only for the Gel-Sight Wedge (Wang et al., 2021) and Mini. Moreover, due to the unavailability of calibration images for Svelte and DenseTact2.0—which are essential for SITR, we omit SITR from the T3 benchmark evaluation.

**Object Classification.** As shown in Table 2, CTSRL significantly outperforms all baselines across both intra-sensor and inter-sensor sets. For challenging transfers between **highly disparate** sensors, such as **DIGIT-Mini**, CTSRL achieves the most substantial lead, improving upon SITR by **nearly 20%**. Figure 3 presents the specific results of SITR and our method on DIGIT-Mini; additional detailed intra- and inter- sensor transfer results comparisons are provided in Section A for better understanding our work. Notably, CTSRL achieves superior performance compared to SITR while using **only 50k** training samples **instead of 1000k**,

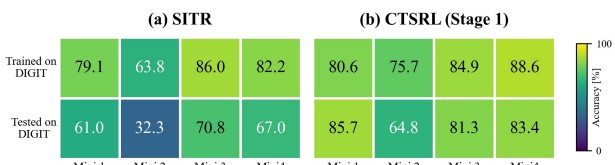

*Figure 3.* Detailed transfer classification results of SITR and CT-SRL on DIGIT-Mini. **Top rows**: trained on DIGIT and tested on GS Mini1-4. **Bottom rows**: trained on GS Mini1-4 and tested on DIGIT, respectively.

showcasing its efficiency in learning transferable tactile representations. This result validates our approach's ability to remove sensor-specific noise and capture purely sensor-invariant features. Even on the T3 bench from Table 3, characterized by extreme sensor variations, CTSRL outperforms T3 despite the latter being equipped with its own specific encoders. Unexpectedly, CTTP exhibits inferior performance compared to the ResNet RI baseline, signaling a pronounced degradation when generalizing to novel, unseen sensor configurations. Section F.1 provides a comprehensive analysis that includes results of CTTP optimized on the same simulation data as Stage 1.

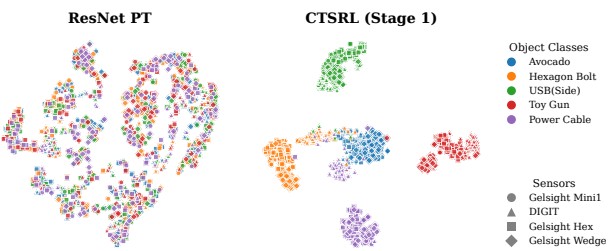

*Figure 4.* t-SNE visualization of the feature space. We compare our results with an ImageNet-pretrained ResNet-50 to demonstrate how our method enhances cluster discriminability.

**Visualization.** To analyze the representation space, we visualize CTSRL features on the SITR Bench using t-SNE (Maaten & Hinton, 2008). As shown in Figure 4, CTSRL produces clear, category-based clusters, whereas ResNet PT fails to do so. This indicates that CTSRL effectively reduces sensor-specific variations and aligns data from different sensors in a shared latent space. Interestingly, the DIGIT sensor exhibits relatively less compact clustering compared to other sensors. We attribute this to its unique optical design, which necessitates the DIGIT-Mini transfer task. This task serves as a more rigorous benchmark for cross-sensor transferability than standard settings.

**Pose Estimation.** For baseline models, we use a similar process to that detailed in Section D and evaluate this task on the inter-sensor dataset. As shown in Table 2, CT-

*Table 4.* Evaluation of tactile perception tasks on seen sensors (GelSight and DIGIT). † indicates datasets seen during pre-training.

| Method | Touch and Go | | | Cloth | Feel | YCB-Slide |
|---|---|---|---|---|---|---|
| | Material | Rough | Hard | Material | Grasp | Object |
| ResNet RI | 37.79 | 78.56 | 76.03 | 20.92 | 65.21 | 46.03 |
| ResNet PT | 53.59 | 84.58 | 85.70 | 25.07 | 70.30 | 63.48 |
| UniTouch | 61.27 | 84.50 | 88.48 | 18.27 | 69.72 | 56.37 |
| AnyTouch | 80.82 | 86.74 | 94.68 | 39.43† | 78.19 | 75.40† |
| AnyTouch$^+$ | 81.39 | 87.48 | 95.65 | 35.63 | 76.42 | 73.96 |
| **Ours (Stage 1)** | 56.42 | 85.32 | 88.02 | 31.06 | 74.76 | 76.48 |
| **Ours (Stage 2)** | **87.70** | **90.58** | **97.21** | **48.33** | **80.28** | **79.11** |

SRL consistently outperforms all baseline models across the SITR benchmark. Notably, on the challenging **DIGIT-Mini** transfer task, CTSRL reduces the RMSE by **approximately 30%** compared to SITR. We also find that ResNet PT significantly improves upon ResNet RI and even surpasses SITR in the DIGIT-Mini scenario. This suggests that for regression-based tasks like pose estimation, knowledge distilled from natural images generalizes effectively to the tactile domain. The results in Table 3 demonstrate that our approach preserves a clear competitive edge, further validating the generalizability of our framework across diverse tactile configurations.

### 4.2. Evaluation on Fine-grained Semantic

**Training Datasets.** For the subsequent **Cross-modal Semantic Alignment** phase, we employ 5 real-world multimodal datasets for training: Touch and Go (TAG) (Yang et al., 2022), TVL (Fu et al., 2024), SSVTP (Kerr et al., 2023), Octopi (Yu et al., 2024), and TacQuad (Feng et al., 2025). These datasets collectively cover four distinct types of visuo-tactile sensors: GelSight, GelSight Mini, DIGIT, and DuraGel (Zhang et al., 2024).

**Benchmarks and Baselines.** To evaluate the semantic discriminability of CTSRL's cross-sensor representations after multi-modal alignment, we conduct linear probing on multiple fine-grained tasks across both seen and unseen sensor domains. Evaluation is conducted on TAG (material/rough/hard), Feel (Calandra et al., 2017), Cloth (Yuan et al., 2018), YCB-Slide (Suresh et al., 2023), and the ObjectFolder (OF) Benchmark (Gao et al., 2021; 2022). The OF Benchmark focuses on material classification and includes the simulated datasets OF 1.0 and OF 2.0, along with OF Real. We compare against recent representative multimodal tactile models: UniTouch and AnyTouch. We also retrain AnyTouch on the multi-sensor dataset used in our Stage 2, denoted as AnyTouch$^+$.

**Evaluation on Seen Sensors.** As shown in Table 4, the introduction of multimodal alignment leads to a significant performance leap over the Stage 1 model, particularly in material classification tasks requiring fine-grained seman-

*Table 5.* Evaluation of material classification on unseen sensors (TACTO, Taxim and GelSlim). † indicates datasets seen during pre-training.

| Method | OF1.0 | OF 2.0 | OF Real |
|---|---|---|---|
| Random Init | 62.36 | 70.77 | 30.40 |
| ImageNet | 66.30 | 72.84 | 39.91 |
| UniTouch | 68.08 | 85.40† | 35.65 |
| AnyTouch | **69.56** | 76.02 | 79.26† |
| AnyTouch$^+$ | 69.14 | 75.70 | 38.78 |
| **Ours (Stage 1)** | 68.89 | **76.78** | 45.60 |
| **Ours (Stage 2)** | 68.74 | 76.08 | **50.14** |

tics. CTSRL after multimodal alignment outperforms both vanilla AnyTouch and AnyTouch$^+$. Although Cloth and YCB-Slide were part of AnyTouch's original training, our model achieves superior results, highlighting the better generalization of our alignment strategy. Notably, even the sim-only Stage 1 model exhibits a consistent improvement over the ResNet PT and maintains robust performance on object-level perception tasks, such as Feel and YCB-Slide. While AnyTouch$^+$ shows improvement over vanilla Any-Touch on the TAG benchmark, we attribute this gain to the increased proportion of TAG samples in its training set.

**Evaluation on Unseen Sensors.** Table 5 presents material classification results. UniTouch's high accuracy on OF 2.0 is due to exposure to that sensor during training, rather than inherent generalization, as seen in its lower performance on the OF Real dataset. Other baselines, however, show comparable performance across both simulation datasets, OF 1.0 and OF 2.0. In contrast, AnyTouch excels on OF Real but performs worse on Cloth and YCB-Slide in Table 4, despite all being seen datasets. We attribute this to the inclusion of the language modality for tripartite alignment in OF Real but not in Cloth and YCB-Slide, which underscores the pivotal role of linguistic information in fine-grained semantic learning. Interestingly, our Stage 1 model performs best on the simulated OF 2.0. We attribute this to the fact that simulated datasets lack the semantic richness of real-world data; thus, further multimodal alignment provides diminishing returns on simulation, whereas our sensor-invariant Stage

1 design already captures the necessary geometric/texture priors. Notably, CTSRL (Stage 2) significantly outperforms AnyTouch$^{+}$ by **over 10%**. Since both use identical data, this confirms the superiority of our alignment strategy.

## 5. Ablation Study

To analyze the contribution of each CTSRL component, we conduct ablation studies across two stages: Cross-sensor Representation Learning (CRL) evaluates the impact of loss normalization, the CSM module, and cross-/intra-sensor losses on object classification of SITR Bench. Cross-modal Semantic Alignment (CSA) examines the stability and efficacy of multimodal alignment by investigating the effects of CRL pre-training, the frozen-anchor vision encoder strategy, and the contribution of individual vision-language modalities across four diverse unseen datasets.

*Table 6.* Ablation study of the Cross-sensor Representation Learning phase on the SITR Benchmark.

| Model | Inter. | DIGIT-Mini | Intra. | No-Trans. |
|---|---|---|---|---|
| **CTSRL(Stage 1)** | **88.52** | **80.62** | **95.00** | 99.58 |
| w/o CSM | $87.52^{\downarrow 1.00}$ | $78.16^{\downarrow 2.46}$ | $94.22^{\downarrow 0.78}$ | $99.56^{\downarrow 0.02}$ |
| w/o cra | $87.09^{\downarrow 1.43}$ | $74.34^{\downarrow 6.28}$ | $92.40^{\downarrow 2.60}$ | $99.27^{\downarrow 0.31}$ |
| w/o ire | $85.30^{\downarrow 3.22}$ | $73.70^{\downarrow 6.92}$ | $92.80^{\downarrow 2.20}$ | $\mathbf{99.62}^{\uparrow 0.04}$ |
| w/o norm. | $49.72^{\downarrow 38.80}$ | $26.02^{\downarrow 54.60}$ | $62.36^{\downarrow 32.64}$ | $90.10^{\downarrow 9.48}$ |
| w/o norm.+cra | $46.13^{\downarrow 42.39}$ | $23.84^{\downarrow 56.78}$ | $62.56^{\downarrow 32.44}$ | $92.96^{\downarrow 6.62}$ |

**Impact of CRL Components.** As shown in Table 6, removing CSM leads to a performance drop, highlighting its efficacy in mitigating cross-sensor bias. Furthermore, excluding the cross-sensor alignment constraint (w/o cra) results in a greater decline, particularly on the DIGIT-Mini setting. This underscores that cross-sensor alignment is vital for learning sensor-agnostic representations and ensuring cross-domain generalization. A more pronounced performance degradation occurs when the intra-sensor loss (w/o ire) is removed, particularly in the Inter-sensor and DIGIT-Mini scenarios. This can be explained by the foundational role of intra-sensor representation in providing a reliable feature base, without which meaningful cross-sensor alignment cannot be effectively ensured. Removing loss normalization (w/o norm.) leads to catastrophic performance collapse. Furthermore, the configuration "w/o norm.+cra"—which is equivalent to performing self-supervised learning strictly according to the official Barlow Twins standard without incorporating cross-sensor alignment—results in a further decline across the Inter. and DIGIT-Mini settings. This ablation indicates that both loss normalization and explicit cross-sensor alignment are indispensable for learning robust and transferable representations

**Effectiveness of CSA Strategies.** From Table 7, the removal of Stage 1 and CSM from CTSRL leads to a con-

*Table 7.* Ablation study of the Cross-modal Semantic Alignment phase across four unseen datasets.

| Model | Cloth (Material) | Feel (Grasp) | YCB (Object) | OF Real (Material) |
|---|---|---|---|---|
| **CTSRL (Stage 2)** | **48.33** | **80.28** | **79.11** | **50.14** |
| w/o CSM | $47.30^{\downarrow 1.03}$ | $78.81^{\downarrow 1.47}$ | $78.99^{\downarrow 0.12}$ | $48.70^{\downarrow 1.44}$ |
| w/o Stage 1 | $47.13^{\downarrow 1.20}$ | $78.33^{\downarrow 1.95}$ | $78.03^{\downarrow 1.08}$ | $46.88^{\downarrow 3.26}$ |
| w/o Freeze VE | $41.98^{\downarrow 6.35}$ | $79.21^{\downarrow 1.07}$ | $78.15^{\downarrow 0.96}$ | $45.74^{\downarrow 4.40}$ |
| w/o Language | $43.20^{\downarrow 5.13}$ | $77.74^{\downarrow 2.54}$ | $77.58^{\downarrow 1.53}$ | $34.09^{\downarrow 16.05}$ |
| w/o Vision | $46.96^{\downarrow 1.37}$ | $79.00^{\downarrow 1.28}$ | $78.09^{\downarrow 1.02}$ | $46.02^{\downarrow 4.12}$ |

sistent performance degradation, primarily on the unseen sensor task (OF Real). This further demonstrates that both strategies are instrumental in enhancing the model's generalization capabilities to novel sensor domains. Notably, the "w/o Freeze VE" strategy (where the **V**ision **E**ncoder is unfrozen) results in a significant performance decline, particularly on material-related tasks such as Cloth and OF Real. This underscores that unfreezing the vision encoder during the alignment phase likely leads to representation drift, thereby confirming the necessity of maintaining the CLIP backbone as a fixed semantic anchor to provide stable multi-modal guidance. A specific discussion on the freezing configuration of the vision encoder in AnyTouch is provided in Section F.2 for further technical comparison. Furthermore, we observe a performance decline when the vision or language modalities are excluded, highlighting the importance of aligning with these modalities to narrow sensor gaps and achieve comprehensive tactile perception. The performance drop is more pronounced when removing language rather than visual modalities, especially in material classification. This observation aligns with our earlier analysis, further corroborating the critical role of the language modality in facilitating fine-grained semantic learning.

## 6. Conclusion

In this paper, we present Cross-Tactile Sensor Representation Learning (CTSRL), a method for learning universal tactile representations across diverse tactile sensors. We introduce a Cross-Sensor Modulator to eliminate sensor-specific biases, together with a two-stage training framework. First, we leverage aligned simulated tactile data and jointly perform cross-sensor representation alignment and intra-sensor representation enhancement to learn sensor-invariant and transferable tactile representations. Subsequently, to compensate for the limited semantic diversity of simulated data, we incorporate real-world multimodal tactile data and perform cross-modal alignment, further enhancing the model's ability to capture fine-grained semantic information. Extensive evaluations demonstrate that CTSRL consistently outperforms existing baselines.

## Acknowledgements

This work was supported in part by the National Natural Science Foundation of China (No. 62306065), part by Fundamental and Interdisciplinary Disciplines Breakthrough Plan of the Ministry of Education of China (No. JYB2025XDXM103), part by the Central Guidance on Local Science and Technology Development Fund of Shanghai City (No. YDZX20253100002004), and also part by the Fundamental Research Funds for the Central Universities.

## Impact Statement

This paper presents work whose goal is to advance the field of Machine Learning. There are many potential societal consequences of our work, none which we feel must be specifically highlighted here.

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

# A. Detail results of SITR Bench

In this section, we present the comprehensive results of downstream sensor transfer experiments conducted on the SITR Bench. Specifically, Figure 5 (a)–(d) illustrate the detailed transfer performance of various baselines under the intra-sensor setting, while (e)–(h) present the results for the inter-sensor setting. Furthermore, Figure 6 (a)–(c) depict the transfer outcomes specifically for the DIGIT-Mini scenario. Finally, Figure 7 (a)–(d) report the detailed error metrics for the pose estimation task in the inter-sensor setting.

### A.1. Object Classification

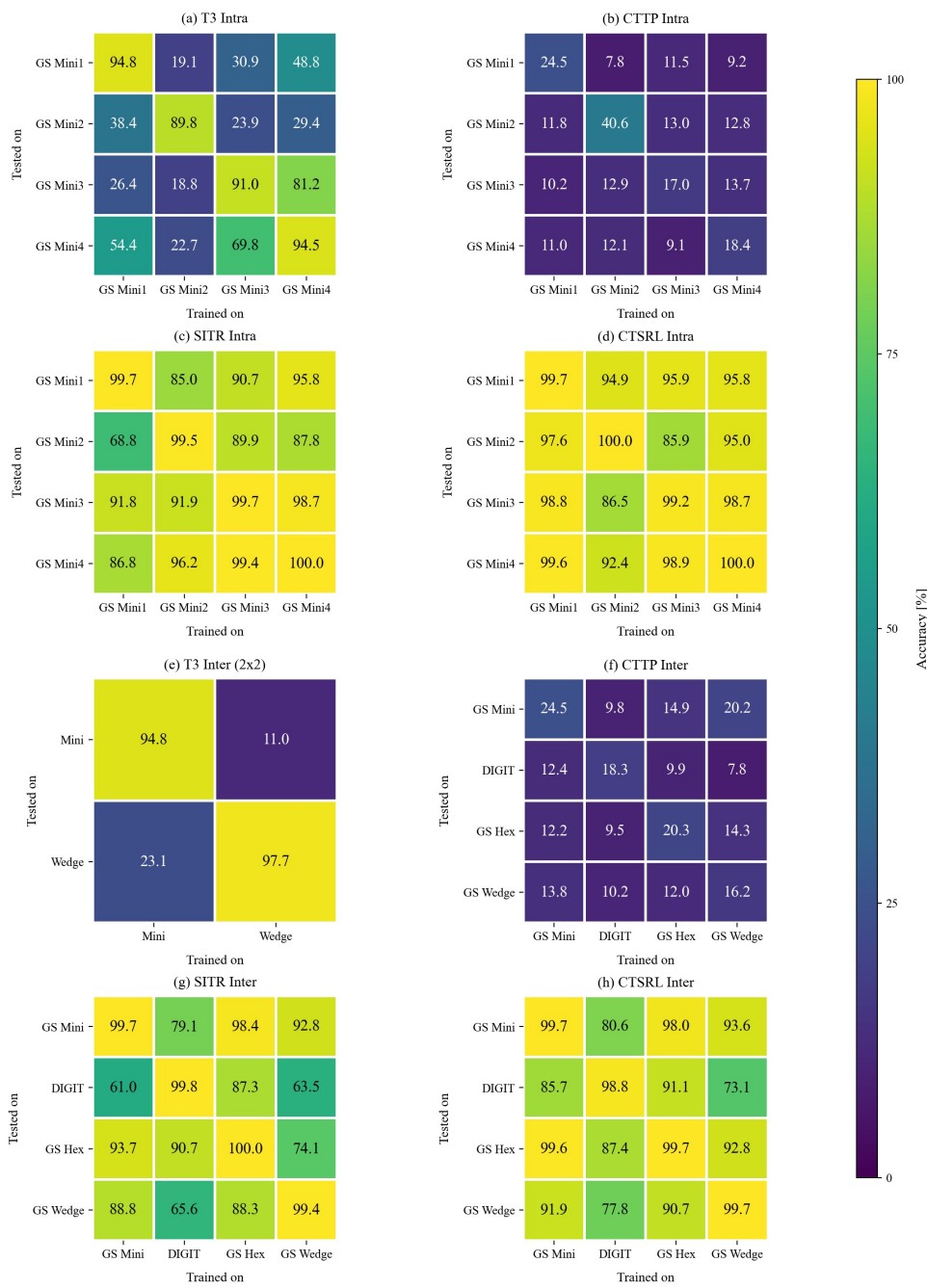

*Figure 5.* Classification on Intra-sensor and Inter-sensor

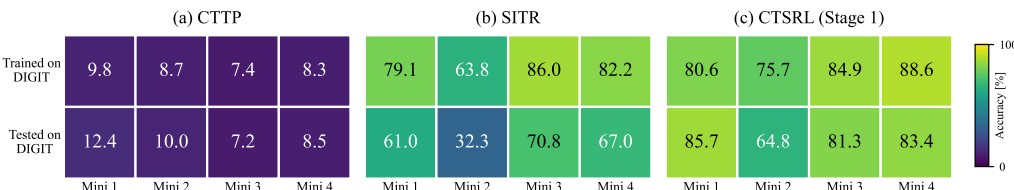

*Figure 6.* Classification on DIGIT-Mini

## A.2. Pose Estimation

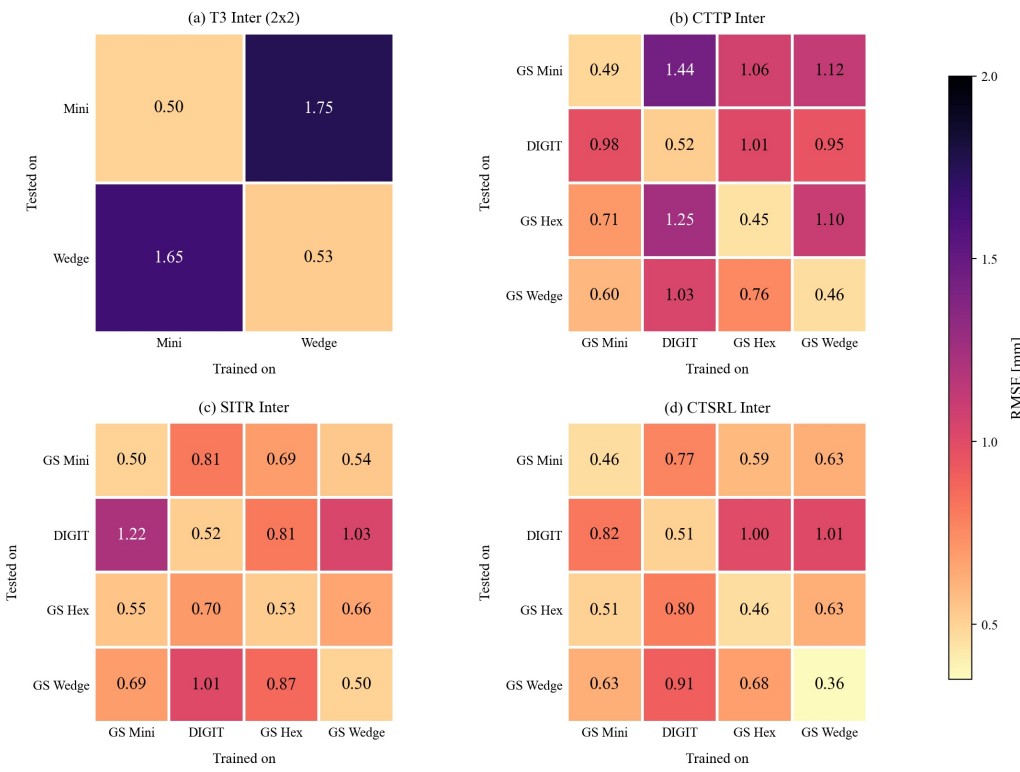

*Figure 7.* Pose Estimation on Inter-sensor

## B. Training Datasets

**Stage 1.** Due to the scarcity of real-world, strictly aligned tactile data, we utilize the open-source simulated data from SITR (Gupta et al., 2025). The simulation allows for precise control over key parameters, including: light properties (shape, orientation, angle, and color), gel properties (stiffness and specularity), and camera properties (Field of View (FOV) and sensing area). For further technical details, please refer to Gupta et al. (2025). We selected the first 5 configurations from this dataset, with each configuration containing 10k images.

**Stage 2.** For the second stage, we incorporated 5 real-world datasets from 4 different sensors for training: Touch and Go (TAG) (Yang et al., 2022) from GelSight (Yuan et al., 2017); TVL (Fu et al., 2024) and SSVTP (Kerr et al., 2023) from DIGIT (Lambeta et al., 2020); Octopi (Yu et al., 2024) from Gelsight Mini (GelSight Inc., 2026); TacQuad (Feng et al., 2025) from DIGIT, GelSight Mini and DuraGel (Zhang et al., 2024). Within these datasets, the language modality in TAG is generated by GPT-4o, and textual descriptions in TVL, SSVTP, and Octopi are extended, as detailed in (Feng et al., 2025). The detailed training dataset statistics are shown in Table 8.

*Table 8.* Training dataset statistics.

| Dataset | Vision | Language | Sensor | Total Size | Used Size |
|---|---|---|---|---|---|
| SITR (Gupta et al., 2025) | ✓ | ✓ | GelSight (Sim) | 1000k | 50k |
| Touch and Go (Yang et al., 2022) | ✓ | ✓ | GelSight | 250k | 250k |
| SSVTP (Kerr et al., 2023) | ✓ | ✓ | DIGIT | 4.5k | 4.5k |
| TVL (Fu et al., 2024) | ✓ | ✓ | DIGIT | 39k | 39k |
| Octopi (Yu et al., 2024) | × | ✓ | GelSight Mini | 39k | 39k |
| TacQuad (Feng et al., 2025) | ✓ | ✓ | GelSight, DIGIT, DuraGel, GS Mini | 55k | 55k |

## C. Downstream Datasets

**SITR Bench.** This benchmark comprises seven distinct tactile sensors, including four GelSight Minis with varying sensor bodies and in-house gel pad modifications, a GelSight Hex (Yuan et al., 2017), a GelSight Wedge (Wang et al., 2021), and a DIGIT sensor. It includes a classification task with 16 objects evaluated across all seven sensors, resulting in a total of 112k tactile images (16k samples per sensor). In addition, a pose estimation task is provided, consisting of 24k tactile images with precise pose annotations, covering six different indenters across four sensors, with 1k samples collected for each indenter–sensor pair.

**T3 Bench.** The T3 Bench (Zhao et al., 2024) is employed to assess zero-shot transfer across four tactile sensors. Specifically, the training set consists of samples from GelSight Wedge and GelSight Mini, whereas the test set comprises data from GelSight Svelte and DenseTact2.0. Detailed information regarding the dataset can be found in (Zhao et al., 2024).

**Benchmark of Fine-grained Semantics Evaluation.** In addition to the primary benchmarks, we selected 7 datasets to evaluate the performance of our model following multi-modal alignment. These include TAG, Feel and Cloth from GelSight, and YCB-Slide (Suresh et al., 2023) from DIGIT. We also incorporated the ObjectFolder (OF) series (Gao et al., 2021; 2023): OF 1.0 and OF 2.0, which utilize Tacto (Wang et al., 2022) and Taxim (Si & Yuan, 2022) simulators respectively, alongside the real-world OF Real data captured by GelSlim (Donlon et al., 2018).

Regarding the experimental protocols, the data partitioning for TAG and OF 2.0 follows the splits established in AnyTouch, while Feel and Cloth adhere to the protocols defined in T3. For OF 1.0 and OF Real, we utilize the official implementations of (Gao et al., 2023). Finally, for the YCB-Slide dataset, we apply a random 8:2 ratio to partition the data into training and test sets.

## D. Model Architecture

**Training Protocol.** Our model utilizes a ResNet-50 as the backbone encoder, complemented by a Cross-Sensor Mapping Module consisting of linear layers and activation functions. In **Stage 1**, the inter- and intra-projectors follow the official Barlow Twins implementation; specifically, the features extracted by the encoder are projected through a three-layer MLP [8192-8192-8192] prior to the computation of the cross-correlation matrix. In **Stage 2**, to achieve multi-modal alignment, we compress the tactile features into a 768-dimensional space to align them with the multimodal feature space of the pre-trained CLIP (Vision-Language) model.

**Classification Decoders.** We employ the Cross-Entropy loss for all classification tasks. For the **SITR** and **T3** baselines, we strictly adhere to their official implementations to ensure consistency. Specifically, for the **SITR Bench**, the class token in **SITR** is projected to 128 dimensions and then down to 16; simultaneously, the patch tokens are processed by a reconstruction decoder to recover the image, which is then fed into a ResNet-18. The resulting feature vector is concatenated with the 16-dimensional class token projection and passed through a 3-layer MLP decoder with dimensions [256, 128, 16]. For **T3**, patch tokens are reconstructed and passed through a ResNet-18 with an output dimension of 16 (which is adjusted to 6 for the **T3 Bench**). For other ResNet-50 based architectures, including **CTSRL**, **CTTP**, **Barlow Twins**, and the vanilla **ResNet-50**, we directly apply a 3-layer MLP decoder with dimensions [256, 128, 16] for SITR Bench (or [256, 128, 6] for T3 Bench) to the output features.

For downstream evaluations after multimodal alignment, to evaluate the quality of the learned representations, we keep the backbone frozen and train only a linear classification head for all models.

**Pose Estimation Decoders.** The Mean Squared Error (MSE) loss is utilized for this task. For **SITR** and **T3**, two tactile

images, $x_1$ and $x_2$, are processed separately through the network. Their corresponding output tokens are reconstructed into images and concatenated along the channel dimension. This joint representation is then passed into a modified ResNet-18 with a 6-channel input, followed by a linear projection to a 3-dimensional output. For the remaining **ResNet-50 based architectures** which lack a reconstruction decoder, we ensure a fair comparison by processing $x_1$ and $x_2$ separately and extracting the feature maps from the third residual layer (with dimensions [512, 14, 14]). Similar to the SITR protocol, these feature maps are concatenated and fed into a modified ResNet-18 with a 6-channel input, before being linearly projected to a 3-dimensional pose vector.

Across all evaluation tasks, only the downstream task decoder is optimized, while the encoder and all other modules remain frozen.

## E. Sensor Embedding Sampling Strategy

To ensure the model effectively learns sensor-agnostic representations while maintaining the ability to calibrate to specific hardware, we employ a dynamic sampling strategy for the sensor embeddings $e$ during different phases of training and evaluation:

- **Cross-sensor Representation Learning.** During the self-supervised learning phase on synthetic data, we set the universal embedding probability $p_u = 1$. Since the sensor-specific "styles" in simulation do not directly translate to real-world hardware, forcing the model to rely exclusively on the universal embedding vector $e_u$ facilitates the acquisition of highly transferable, structural tactile representations.

- **Cross-modal Semantic Alignment.** When transitioning to real-world multimodal datasets encompassing diverse sensor types (*e.g.*, GelSight, GelSight Mini, DIGIT, and DuraGel), we adopt a mixed sampling strategy. The probability $p_u$ is linearly scheduled to increase from 0 to $p_{max}$ (e.g., 0.75). This incremental shift encourages the model to decouple hardware-induced biases from intrinsic physical signals, effectively promoting the learning of hardware-agnostic semantic features.

- **Inference Stage.** For seen sensors, we employ $p_u = 0$ and utilize their corresponding learned sensor-specific embeddings to leverage precise hardware-distinct characteristics and maximize performance. For all downstream evaluations and when encountering unseen sensors, we consistently employ $p_u = 1$. By performing calibration via the universal sensor embedding $e_u$, the model maintains robust feature extraction capabilities without requiring prior knowledge.

## F. Additional Ablation

### F.1. Impact of Self-Supervised Objectives on Cross-sensor Transfer Performance

In this section, we investigate the impact of the contrastive learning objective used in CTTP on model performance. Since the original CTTP does not utilize pre-trained weights from ImageNet, we initialize a ResNet-50 with ImageNet pre-training and apply contrastive learning between aligned tactile images. Specifically, we use the same dataset from the Cross-sensor Representation Learning phase, sampling two images from each of the two different sensor configurations at each step—resulting in four images that form two pairwise aligned pairs—to train the CTTP$^+$ model. All other hyperparameters and settings follow the original CTTP configuration. In essence, CTTP$^+$ represents an enhanced implementation of the CTTP methodology, optimized through our synchronized data and standard pre-trained initializations.

*Table 9.* Comparison of CTTP$^+$ and CTSRL performance on SITR Bench

| Model | Object Classification (Acc % ↑) | | | | Pose Estimation (RMSE mm ↓) | | |
|---|---|---|---|---|---|---|---|
| | Inter-sensor | DIGIT-Mini | Intra-sensor | No-Transfer | Inter-sensor | DIGIT-Mini | No Transfer |
| **CTSRL** | **88.52**±8.01 | **80.62**±7.45 | **95.00**±4.62 | 99.58±0.45 | **0.75**±0.15 | **0.79**±0.04 | **0.45**±0.06 |
| ResNet PT | 73.03±14.58 | 50.31±11.05 | 75.44±17.55 | 96.48±3.94 | 0.81±0.19 | 0.92±0.10 | 0.48±0.02 |
| CTTP | 12.24±3.13 | 9.03±1.67 | 11.25±1.85 | 22.18±8.55 | 1.00±0.23 | 1.21±0.33 | 0.48±0.03 |
| CTTP$^+$ | 72.04±20.39 | 59.43±9.09 | 88.02±9.55 | 99.01±0.90 | 0.76±0.18 | 0.88±0.12 | 0.45±0.04 |

*Table 10.* Ablation Study of Vision Encoder Adaptation Strategy for AnyTouch.

| Method | Strategy | TAG (Material) | Cloth (Material) | Feel (Grasp) | YCB (Object) | OF Real (Material) |
|--------|----------|----------------|------------------|--------------|--------------|--------------------|
| AnyTouch | w/o Freeze Vision Encoder | 80.82 | 39.43† | 78.19 | 75.40† | 79.26† |
| AnyTouch$^+$ | w/o Freeze Vision Encoder | 81.39 | 35.63 | 76.42 | 73.96 | 38.78 |
|  | w Freeze Vision Encoder | 88.83 | 43.94 | 78.76 | 80.03 | 42.90 |

As shown in Table 9, CTTP$^+$ exhibits a significant performance gain over the original CTTP, suggesting that ImageNet pre-trained weights provide a beneficial initialization for tactile representations. Furthermore, compared to the vanilla ImageNet pre-trained ResNet-50, CTTP$^+$ shows substantial improvements across all scenarios except for the Inter-sensor setting. However, even with these enhancements, CTTP$^+$ remains considerably behind our proposed CTSRL in sensor transfer tasks. Specifically, in object classification, CTTP$^+$ lags by approximately 20% on the DIGIT-Mini task and 16% on the Inter-sensor task. Regarding pose estimation, although the performance of the two models appears closer in simpler scenarios, our CTSRL still significantly outperforms CTTP$^+$ in the more challenging DIGIT-Mini transfer task. While CTTP$^+$ achieves an error of 0.88 mm in this scenario, CTSRL reduces this to 0.79 mm, demonstrating better precision and robustness. This fair comparison rigorously demonstrates the superiority of our proposed methodology over existing contrastive learning frameworks in bridging the tactile domain gap, highlighting the benefits of CTSRL's architecture and training pipeline.

### F.2. Effect of Frozen Vision Encoder in AnyTouch

This subsection examines the effect of freezing the CLIP vision encoder within the AnyTouch$^+$ framework. We compare the performance of AnyTouch$^+$ with frozen and unfrozen vision backbones under the same training protocol, while keeping the text encoder fixed to ensure a consistent linguistic reference. As shown in Table 10, freezing the vision encoder leads to consistent performance improvements across all seen and unseen benchmarks. These results suggest that maintaining a fixed visual representation can improve the stability of multimodal semantic alignment in existing methods.

## G. Limitations

Despite the promising results demonstrated by our proposed cross-modal tactile-visual representation learning framework, several key limitations remain to be addressed in future work. First, the pre-training stage (Stage 1) currently relies exclusively on synthetic simulation data, which suffers from insufficient diversity and lacks critical real-world physical properties, including fine-grained surface textures, material reflectance variations, and realistic contact deformation patterns. This domain gap may restrict the model's generalization ability when deployed in unconstrained real-world scenarios. Second, our current CTSRL model is designed for single-frame input and cannot explicitly model the temporal dynamics of tactile signals, such as contact sliding, continuous force variations, and transient tactile events during manipulation. These dynamic cues are essential for achieving robust and dexterous robotic manipulation capabilities. Third, all experiments in this work are conducted on pre-collected offline datasets, and we have not yet performed systematic validation on real-robot manipulation tasks. Real-world challenges such as sensor noise, actuator latency, and environmental uncertainties have not been fully evaluated, which is a critical step toward practical deployment.

Building upon these limitations, our future research will focus on three core directions: (1) constructing a large-scale multi-modal tactile-visual dataset that combines high-fidelity simulation data with real-world tactile measurements across diverse materials and textures; (2) extending the CTSRL framework to incorporate temporal modeling modules (e.g., temporal convolutions or transformers) to capture dynamic tactile signal variations; and (3) validating the proposed method on a suite of real-robot manipulation tasks including grasping, insertion, and surface inspection. In parallel, we will also explore the broader applicability of this cross-modal representation learning paradigm to other domains beyond robotics, such as remote sensing and medical imaging.

