# OpenReview forum: "Cross-Tactile Sensor Representation Learning"
_ICML.cc/2026/Conference — ICML 2026 regular_

### Official Review · Reviewer_FPj9 · 2026-03-07

**Soundness:** 3
**Presentation:** 3
**Significance:** 3
**Originality:** 3
**Overall Recommendation:** 5
**Confidence:** 2

**Summary:**

The paper presents a novel method for creating tactile sensor representations that are agnostic to the hardware and other factors like noise. Authors conduct evaluation on the real and synthetic datasets, and show improved performance in comparison to previous approaches.

**Compliance With Llm Reviewing Policy:**

Affirmed.

**Final Justification:**

My questions and concerns were fully addressed. I keep my score as is, since it is already accept, and I didn't see anything to change my mind. It is interesting and technically solid work.

**Key Questions For Authors:**

What is the rationale for using ResNet-50 as a tactile encoder vs. ViT models ?

Was there any dynamic component to the approach (E.g, using multiple consecutive tactile images instead of just one). It seems like a lot of useful information is encoded in the dynamic changes. If using a single tactile image, how was it determined what specific image to use (from a time sequence)

What kind of augmentations were applied to the synthetic dataset to simulate different hardware and noise scenarios?

**Limitations:**

The authors do not discuss limitations of this work. It would be good to add a limitations section in the paper.

**Strengths And Weaknesses:**

***Soundness***
Overall, the paper appears to be technically sound in both engineering approach and the evaluations. It is convincing that the approach works.
The performance and limitations are clearly understood from the experiments. The evaluation datasets and baseline models seem to be sound. The evaluations are also done on unseen real sets, so we learn about the generalization of the approach

Some comments and suggestions:

Is the ResNet-50 tactile encoder frozen or is trained ? Is it just loaded with the original weights and further trained or fine-tuned? It could be good to explain

It would be good to explain why in Table 3, CTTP baseline model was even worse than ResNet RI. I would think that a tactile model would be better than randomly initialized ResNet. Could it be an issue with the baseline implementation.

Are there any specific sensors or objects where the model failed to generalize on the unseen data? For example, does the model fail more on soft or rounded objects ?  It would be good to explore some of the cases where the approach did not work well. This might be something to add to the appendix.

***Presentation***
Overall the presentation is good. The paper is easy to understand and follow. The results are well presented and enough details are given to reproduce the work.
However, the presentation could be improved.
Few comments:
What is inside the “Modulator” in FIgure 2. Could be good to show the internal architecture on the figure.

It could be good to discuss limitations and future work for this area. Some things that come to mind are: modeling of dynamic changes or evaluation on downstream tasks beyond recognition, like using the tactile representations to improve manipulation

In equation 5, what coefficients were used for training?

***Significance***
Creating robust and hardware representations of the various tactile sensors is an important and relevant problem for robotics. The paper achieves a new state-of-the-art on many benchmarks, so it's a good contribution.
Also, the impact of such representations can be broadened beyond robotics, as it is a common problem of learning to adapt to understand different configurations of sensor hardware. For example, sensor arrays like microphone and radar arrays have very diverse configurations, which are difficult to analyze. I would be curious to see if the approach in the paper would work there.

***Originality***
The topic of hardware agnostic tactile sensors have been explored before, as the authors provide some previous work on the general idea. The main contribution of the paper is the approach, which uses a novel Cross-Sensor Modulator. As I understand, this approach has not been applied before for hardware agnostic tactile representations. While the CSM doesn’t make a huge difference in the ablation study, it seems that the components of the system work together to get it to state-of-the-art.

---

> ### Author Rebuttal · Authors · 2026-03-30
>
> We sincerely thank the reviewer for the positive and constructive feedback. We address all your comments and questions point-by-point below, with corresponding revisions to be made in the updated manuscript.
>
> # 1. Method & Experiment Details
> ## Q1: Training status of the ResNet-50 tactile encoder
> We clarify the ResNet-50 encoder protocol:
> - **Initialization**: with official ImageNet pre-trained weights;
> - **Training**: the encoder and CSM are fully trainable in both Stage 1 and 2;
> - **Evaluation**: the CTSRL encoder is frozen, with only task-specific heads trained (details in Appendix D).
>
> We will add this clarification to Appendix D in the revised manuscript for full reproducibility.
>
> ## Q2: CTTP baseline underperformance
> We use the official CTTP weights. Its poor performance is mainly because it is trained on a randomly initialized ResNet-50 with data from Soft Bubble and GelSlim sensors, which have a significant domain gap with the sensors in our downstream evaluation. For a fair comparison, we re-implemented CTTP with an ImageNet-pretrained ResNet-50 and the same training data as CTSRL Stage 1 (denoted as CTTP+). As detailed in Appendix F.1, CTTP+ outperforms the original CTTP but still underperforms CTSRL, verifying the superiority of our method.
>
> ## Q3: Failure cases
>
> Selected failure examples are shown at https://anonymous.4open.science/r/ICML-64D6/failure%20cases.jpg; a more comprehensive analysis will be included in the revised appendix.
>
> # 2. Presentation Improvements
> ## Q1: "Modulator" in Figure 2
>
> The mathematical formulation of the CSM is detailed in Section 3.1. To improve clarity, we will update Figure 2 to include a detailed internal schematic of the CSM in the revised manuscript.
>
> ## Q2: Coefficients in Equation 5
>
> For Equation 5, we use $λ^{cra}​=1.0$ (cross-sensor alignment loss) and $λ^{ire}​=0.5$ (intra-sensor enhancement loss), selected via grid search to balance feature alignment and discriminability and prevent representation collapse. We will explicitly state these values in Section 3.2 of the revised manuscript.
>
> ## Q3: Limitations and future work discussion
>
> Due to space constraints, we did not elaborate on these aspects in the paper. As you noted, our core limitations are:
>  - The model currently lacks explicit modeling of dynamic tactile signal variations, such as contact sliding and force control；
> - The method lacks validation on real-robot manipulation tasks. These will all be the core focus of our future work.
>
> Moreover, we share your view that this framework has potential beyond robotics, which aligns perfectly with our research vision. We will extend this framework to other fields including remote sensing and medical imaging in future research.
>
> We will add a dedicated **Limitations and Future Work** section to systematically discuss the above content in the revised manuscript.
>
> # 3. Key Questions
> ## Q1: Rationale for ResNet-50 over ViT as tactile encoder
>
> Our choice of ResNet-50 is motivated by two key factors: (1) **deployment efficiency**, as real-world robotic systems are often resource-constrained, where ResNet-50 offers substantially lower computational and memory overhead than ViT-Base/Large (see **Q2 from Reviewer kp72**), enabling real-time inference; and (2) **generalization**, as CTSRL consistently achieves strong performance on ViT-Small/Base (see **Q1 from Reviewer 2TbY**), demonstrating that our approach is not tied to a specific backbone and readily extends to Transformer architectures.
>
> ## Q2: Dynamic component and single-frame selection
>
> Our CTSRL framework follows the same input setting as SITR (Gupta et al., 2025), operating on single-frame tactile images without any dynamic temporal component (which we will explicitly discuss as a limitation in the revised manuscript). All datasets used in this work provide static tactile images captured at stable sensor–object contact, and we directly adopt these official frames without any additional selection from temporal sequences.
>
> This design ensures a fair and consistent comparison with prior methods under the same single-frame protocol. As noted in our future work, we plan to extend CTSRL to sequential tactile data and incorporate temporal modeling in future research.
> ## Q3: Synthetic dataset augmentation strategy
>
> We adopt the open-source, physics-based rendered synthetic tactile dataset from SITR. As elaborated in the DATASETS section of the original SITR paper, this dataset natively parameterizes core hardware attributes (including illumination, gel, and camera properties) across 100 unique sensor configurations to simulate the heterogeneity of visuo-tactile sensors.
>
> We further apply standard augmentations including random flipping, color jitter, Gaussian blur and noise injection during training to boost the model’s robustness to real-world sensor noise and imaging variations. This detailed strategy will be added to the Training Datasets subsection of the revised manuscript for full reproducibility.

---

> > ### Author Rebuttal · Reviewer_FPj9 · 2026-04-01
> >
> > I thank the authors for a detailed rebuttal.
> > The rebuttal addresses my questions, my opinion (Accept) on the paper remains unchanged.

---

> > > ### Author Response · Authors · 2026-04-02
> > >
> > > We sincerely thank Reviewer FPj9 for your careful review, valuable feedback and positive recognition of our work. We are pleased that our responses have fully addressed your concerns, and we will integrate all supplementary content and clarifications from the rebuttal into the final manuscript. Thank you again for your support.

---

### Official Review · Reviewer_2TbY · 2026-03-13

**Soundness:** 2
**Presentation:** 3
**Significance:** 2
**Originality:** 2
**Overall Recommendation:** 4
**Confidence:** 3

**Summary:**

This work focus on the learning of sensor-invariant representations between heterogeneous visuo-tactile sensors(e.g. GelSight, DIGIT, etc.). They propose Cross-Tactile Sensor Representation Learning (CTSRL), a novel framework, built upon it's Cross-Sensor Modulator. The approach employs a two-stage training paradigm: first achieves structural alignment through cross-sensor representation learning, and subsequently performs semantic enrichment via multi-modal alignment. They conduct experiments on SITR Bench, T3 Bench and several fine-grained semantic benchmarks. The results show that CTSRL achieves SOTA performence compared to other baselines(e.g. SITR, AnyTouch, etc.), even with smaller model size and less training data.

**Compliance With Llm Reviewing Policy:**

Affirmed.

**Final Justification:**

Thanks for the authors' response. The rebuttal has addressed my concerns. Therefore, I am willing to raise my score. I suggest the authors add these points to the final version of the paper to improve it.

**Key Questions For Authors:**

see weakness

**Limitations:**

Yes

**Strengths And Weaknesses:**

Strengths
1. This work propose to address the transferable representation learning across different visuo-tactile sensors, which is the bottleneck of robotic manipulation, and the experiments results show that CTSRL can learn transferable representations between different sensors. Furthermore, the proposed two-stage framework, incorporating cross-sensor and intra-sensor losses, represents a novel contribution to the field of visuo-tactile sensing.
2. This work provide comprehensive ablation experiments to varify the effectiveness of each part of CTSRL, especially in the DIGIT-Mini transfer task, the results show pronounced performance degradation occurs when the core components (i.e., CSM and ire, etc) are removed.
3. The proposed CTSRL is parameter-efficient, which has advantages over other baselines(e.g. AnyTouch, UniTouch, etc.), and they can achieve better performance with less training data, which is more practical in transfer setting for deployment.
4. The paper is well presented, the related work offers a well-structured taxonomy of prior work, and the pipeline in fig.2 provides a clear illustration of the data flow of their framework. The experimental evaluation is comprehensive, spanning both geometric and semantic tasks while including experiments on both seen and unseen sensors.

Weaknesses:
1. ResNet50 is used as the base model in this work, however, comparative experiments involving alternative architectures are missing to further validate the effectiveness of the proposed CTSRL.
2. Lacking real-world robot manipulation experiments, the fundamental value of tactile representation learning lies in supporting robot manipulation tasks. However, the paper only conducted offline benchmark tasks such as classification and pose estimation, without validating them in real-world robot manipulation scenarios.
3. Regarding the evaluation metric used in Table 2, the authors should provide more details on the evaluation metric used in the experiments for clarity.
4. CTSRL's Stage I relies entirely on synthetic data generated by the TACCHI simulator for pre-training, while Stage II still requires collecting labeled real data for each downstream task for fine-tuning. This weakens the practical value of synthetic pre-training.
5. Lack of comparison with more baseline methods: The paper lacks comparison with more extensive cross-domain transfer learning methods (such as classic methods of domain adaptation and domain generalization), making it difficult to determine the real advantages of CTSRL's two-stage design compared with general transfer learning paradigms.

---

> ### Author Rebuttal · Authors · 2026-03-30
>
> We thank the reviewer for the constructive comments. We address each concern below with supplementary experiments and clarifications, while all core results and conclusions remain unchanged.
> # Q1: Backbone Generalization
> We choose ResNet-50 due to its small parameter size and high inference efficiency (see **Q2 from Reviewer kp72**), but our method is not architecture-specific. To demonstrate this architectural generality, we evaluate two widely used transformer backbones, ViT-Small (ViT-S) and ViT-Base (ViT-B), on the SITR Bench.
>
> Table 1. CTSRL Generalization on Different Backbones (SITR Bench, Object Classification)
>
> |Model|Intra.|Inter.|DIGIT-Mini|
> |-|:-:|:-:|:-:|
> |ViT-S(ImageNet)|91.31|75.74|84.57|
> |**ViT-S(CTSRL)**|**94.03**|**84.48**|**88.62**|
> |ViT-B(ImageNet)|94.62|89.13|89.38|
> |**ViT-B(CTSRL)**|**98.24**|**91.39**|**94.65**|
>
> Results show CTSRL consistently improves cross-sensor generalization across both backbones, outperforming ImageNet-pretrained baselines on all metrics. Notably, on the challenging DIGIT-Mini task, CTSRL yields absolute gains of 4.05% (ViT-S) and 5.27% (ViT-B). This confirms that our core innovations (Cross-Sensor Modulator and synergistic learning) are backbone-agnostic and not limited to ResNet-50.
> # Q2: Real-World Robot Manipulation Experiments
> We fully agree on the critical value of real-world closed-loop manipulation experiments for tactile representations. Due to space constraints, please refer to **Q1 from Reviewer kp72**.
> # Q3: Evaluation Metric Details for Table 2
> We apologize for the omission. For Table 2, the metrics are:
> - **Object Classification:** Top-1 transfer accuracy (%).
> - **Pose Estimation:** RMSE (mm) of 3-DoF positional changes from paired tactile frames.
>
> All values represent the mean and standard deviation across the specified sensor transfer sets. We will add this metric clarification in the revised manuscript.
> # Q4: Practical Value of Synthetic Pre-Training
>
> ## Factual Clarifications
> 1. Stage 1 synthetic data is sourced from the open-source SITR dataset, not TACCHI.
> 2. Stage 2 performs **unsupervised cross-modal alignment without any task-specific labels**. For all downstream evaluations, the CTSRL encoder is strictly frozen; only the lightweight task head is trained. This fundamentally avoids the need for large-scale labeled real data required by full-model fine-tuning.
> ## Practical Effectiveness of Synthetic Pre-Training
>
> The synthetic pre-training in Stage 1 forms the core of our framework and alone provides **an efficient, sensor-agnostic tactile representation solution**, without any real data adaptation:
>
> 1.  **High Efficiency & Zero-Shot Capability:** Using only 50k synthetic samples (5% of SITR), Stage 1 outperforms prior SOTAs (T3, SITR) on zero-shot cross-sensor object classification and pose estimation.
> 2.  **Unseen-Sensor Generalization (Table 5):** On the unseen-sensor material classification task, even without cross-modal alignment, Stage 1 independently achieves 76.78% on OF 2.0 (outperforming all baselines) and surpasses AnyTouch+ by 6.82% on OF Real (second only to our Stage 2 model).
> 3.  **Crucial Ablation (Table 7):** Removing Stage 1 causes consistent performance drops, including a 3.26% decline on OF Real.
>
> Stage 2 is only a complementary semantic enhancement module to ensure fair comparison with multimodal baselines (UniTouch, AnyTouch), and is **not a prerequisite** for the effectiveness and practical value of synthetic pre-training.
> # Q5: Comparison with Domain Adaptation Baselines
> For a fair comparison, we evaluated classic domain adaptation (DA) baselines: DANN (Ganin et al., 2016) and Deep CORAL (Sun & Saenko, 2016), all using the **same ResNet-50 backbone, intra-sensor Barlow Twins pre-training, and training data** as CTSRL Stage 1, with only the cross-domain module (consisting of inter-sensor alignment and cross-sensor modulator) replaced.
>
> Table 2. CTSRL vs. Domain Adaptation Baselines (SITR Bench, Object Classification)
>
> |Model|Intra.|Inter.|DIGIT-Mini|
> |-|:-:|:-:|:-:|
> |Barlow Twins|92.40|87.09|74.34|
> |DANN|93.24|85.70|71.70|
> |CORAL|88.68|82.41|64.82|
> |**CTSRL**|**95.00**|**88.52**|**80.62**|
>
> Results show CTSRL consistently outperforms all baselines across all transfer scenarios, especially on the challenging DIGIT-Mini task, with an 8.92% accuracy gain over DANN and 15.80% over CORAL.
>
> Notably, classic DA methods even underperform the Barlow Twins baseline in inter-sensor and DIGIT-Mini transfer. Their rigid global distribution alignment fails to disentangle sensor-specific hardware biases from invariant physical semantics, leading to suboptimal feature transfer in heterogeneous tactile settings and degrading discriminative pre-trained features.
>
> In contrast, CTSRL’s Cross-Sensor Modulator eliminates sensor biases via adaptive modulation and preserves semantic features through synergistic cross/intra-sensor learning, making it better suited for cross-sensor tactile transfer and superior to general DA paradigms.

---

> > ### Author Rebuttal · Reviewer_2TbY · 2026-04-03
> >
> > Thanks for the authors' response. The rebuttal has addressed my concerns. Therefore, I am willing to raise my score. I suggest the authors add these points to the final version of the paper to improve it.

---

> > > ### Author Response · Authors · 2026-04-03
> > >
> > > We sincerely thank Reviewer 2TbY for the positive feedback and valuable suggestions. We will incorporate the relevant points into the final version and greatly appreciate the reviewer’s time and guidance.

---

### Official Review · Reviewer_kp72 · 2026-03-26

**Soundness:** 3
**Presentation:** 3
**Significance:** 2
**Originality:** 3
**Overall Recommendation:** 4
**Confidence:** 4

**Summary:**

This work focuses on the severe heterogeneity across visuo-tactile sensors caused by differences in their designs, which makes it difficult for models to generalize across sensors. To address this issue, the authors propose the CTSRL framework, introducing a Cross-Sensor Modulator (CSM) to remove sensor-specific biases and jointly leverage cross-sensor and intra-sensor learning to extract shared tactile representations. The method first performs self-supervised learning on aligned simulated data to achieve structural alignment across sensors. It then incorporates real-world tactile-vision-language multimodal data for semantic alignment, bridging the gap between simulation and reality. Ultimately, the approach learns sensor-agnostic representations with strong generalization ability and rich fine-grained physical semantics.

**Compliance With Llm Reviewing Policy:**

Affirmed.

**Key Questions For Authors:**

See the Strengths and Weaknesses part.

**Limitations:**

yes

**Strengths And Weaknesses:**

## Strengths
1. The paper is well-written, with a clear and intuitive presentation of the proposed method.
2. The experimental evaluation is thorough, including comparisons with strong and recent baselines, providing solid empirical evidence.
3. The work is well-motivated from both practical and experimental perspectives.
4. The method generalizes effectively to unseen sensors, demonstrating strong robustness and real-world applicability.

## Weaknesses
1. The paper lacks real-world (physical) experiments, which limits the validation of the method’s effectiveness and its practical applicability in real robotic settings.
2. Although Table 1 highlights that the method has fewer parameters than ViT-based approaches, the paper does not provide inference time comparisons; including such results would better demonstrate efficiency advantages.

---

> ### Author Rebuttal · Authors · 2026-03-30
>
> We sincerely thank the reviewer for the positive and constructive feedback. We address all your comments and questions point-by-point below, with corresponding revisions to be made in the updated manuscript.
> # Q1: Real-World Robot Manipulation Experiments
> Our research focuses on solving the key bottleneck limiting widespread tactile perception in manipulation tasks: **the lack of general, sensor-agnostic tactile representations with cross-sensor generalization ability**. Without solving this fundamental problem, manipulation policies can only be tied to a specific tactile sensor and fail to generalize to diverse real-world tactile hardware, severely limiting the practical deployment of tactile perception. The core goal of this work is thus to tackle the **cross-sensor tactile representation learning** problem. Following prior work SITR (which also **did not conduct real-robot manipulation experiments**), we evaluate representations via standardized offline benchmarks. These tasks can systematically and quantitatively assess the core metrics of tactile representations: feature discriminability and cross-sensor generalization ability, the core prerequisite for real-robot deployment.
>
> We fully agree that the ultimate value of tactile representation learning is to enable real-world robotic manipulation. Our lab has a Franka Panda manipulator (https://anonymous.4open.science/r/ICML-64D6/franka.jpg), and we purchased two tactile sensors (GS Mini1 and DIGIT https://anonymous.4open.science/r/ICML-64D6/sensor.jpg) upon receiving this comment. However, due to the tight rebuttal timeline, custom mechanical connectors for sensor-manipulator integration are not yet available, so we cannot complete the full tactile-enabled real-robot manipulation experiment within this period. For this reason, building on our prior validation using third-party public tactile datasets, we performed supplementary verification with real tactile data independently collected by the two sensors, to further validate the real-world cross-sensor generalization performance of our method.
> ## Supplementary Experiment
> Following the SITR protocol, we collect real tactile data of two typical objects (avocado and lemon) with GS Mini1 and DIGIT as the held-out test set (https://anonymous.4open.science/r/ICML-64D6/data.jpg). The encoder is frozen, and only a 16-class classification head is trained on the SITR dataset and evaluated on our self-collected test set, using argmax over the full 16-class output space (without restricting to the two test categories).
>
> Table 1. Zero-shot cross-sensor classification accuracy on our self-collected real test set (**SITR**)
>
> |Test\Train|GS Mini1|GS Mini2|GS Mini3|GS Mini4|DIGIT|GS Hex|GS Wedge|
> |-|:-:|:-:|:-:|:-:|:-:|:-:|:-:|
> |GS Mini1|-|8.75|35.75|20.50|22.25|51.00|20.25|
> |DIGIT|56.75|6.00|50.75|12.25|-|36.25|6.50|
>
> Table 2. Zero-shot cross-sensor classification accuracy on our self-collected real test set (**CTSRL**)
>
> |Test\Train|GS Mini1|GS Mini2|GS Mini3|GS Mini4|DIGIT|GS Hex|GS Wedge|
> |-|:-:|:-:|:-:|:-:|:-:|:-:|:-:|
> |GS Mini1|-|38.40|93.54|93.16|61.98|96.96|41.63|
> |DIGIT|77.75|61.75|97.00|94.00|-|81.00|71.00|
>
> Results show that CTSRL significantly outperforms SITR on completely unseen real-world test data collected by heterogeneous tactile sensors. Specifically, in cross-sensor testing scenarios, CTSRL maintains strong feature discriminability and generalization performance, while the performance of SITR drops sharply. This supplementary experiment fully demonstrates that the general tactile representation learned by our method has excellent zero-shot generalization ability on heterogeneous tactile sensors in the real world, which lays a solid core foundation for subsequent deployment in closed-loop real robot manipulation tasks.
>
> We will include these results in the final version. Meanwhile, we will complete the hardware integration and conduct full real-robot manipulation experiments to further evaluate the end-to-end performance of our method. These directions will be discussed in the future work section.
> # Q2: Inference Efficiency Comparison
> We evaluate inference efficiency under a unified setting: a single NVIDIA RTX 5090 GPU, PyTorch/CUDA environment, 224×224 input, and batch size = 1. Each model is measured after 100 warm-up iterations, followed by 1000 inference runs, with results averaged.
>
> Table 3. Inference Efficiency Comparison
>
> _SFIT: Single-Frame Inference Time; Throughput: Frames Per Second (FPS)_
>
> |Method|BaseModel|Params.|SFIT(ms,↓)|Throughput(FPS,↑)|
> |-|:-:|:-:|:-:|:-:|
> |ResNet PT|ResNet-50|25.6M|3.78|264.36|
> |CTSRL|ResNet-50|26.1M|3.87|258.53|
> |SITR|ViT-B|96.0M|5.84|171.33|
> |AnyTouch|ViT-L|304.0M|25.41|39.35|
>
> CTSRL achieves comparable efficiency to the ResNet-50 baseline, while delivering **1.5×** and **6.6×** speedups over SITR and AnyTouch, respectively. These demonstrate that CTSRL provides strong generalization without sacrificing real-time performance.

---

> > ### Author Rebuttal · Reviewer_kp72 · 2026-04-03
> >
> > Thanks to the author for the reply—my question has been well resolved.

---

> > > ### Author Response · Authors · 2026-04-03
> > >
> > > We sincerely thank Reviewer kp72 for your positive acknowledgement and valuable feedback. We greatly appreciate your time and careful effort in reviewing both our manuscript and our rebuttal.

---

### Decision · Program_Chairs · 2026-04-30

**Decision:**

Accept (regular)

**Comment:**

This paper proposes a transferable visuo-tactile representation learning approach. Reviewers praise its clear writing, strong ablations, parameter efficiency, and SOTA results on benchmarks. However, major weaknesses persist: more real-world robot manipulation experiments, synthetic pre-training still requires task-specific real data for fine-tuning, missing comparisons with domain adaptation/generalization baselines, and no analysis of failure cases. But the strengths overrides the weaknesses, so I am inclined toward acceptance.